# Comparison of the Three-Phase Corrosion Behavior of SiN and 304L Stainless Steels in 6 M Nitric Acid Solution at Different Temperatures

Shengxuan Sun [1,2], Lianmin Zhang [1,*], Aili Ma [1], Enobong Felix Daniel [1], Chunzhi Zhang [2,*] and Yugui Zheng [1]

1   CAS Key Laboratory of Nuclear Materials and Safety Assessment, Institute of Metal Research, Chinese Academy of Sciences, 62 Wencui Road, Shenyang 110016, China; 17864288820@163.com (S.S.); alma@imr.ac.cn (A.M.); enobong18b@imr.ac.cn (E.F.D.); ygzheng@imr.ac.cn (Y.Z.)
2   School of Materials Science and Engineering, Shandong University of Science and Technology, Qingdao 266590, China
*   Correspondence: lmzhang14s@imr.ac.cn (L.Z.); czzhang@sdust.edu.cn (C.Z.)

**Abstract:** In this work, the three-phase corrosion behavior of SiN and 304L stainless steels was comparatively investigated in a 6 M nitric acid solution at different temperatures. It was found that the corrosion rates of both steels in the liquid phase, vapor phase and condensate phase of nitric acid showed an increasing trend with rising temperature. Meanwhile, there also existed some differences in the corrosion kinetics and the corrosion resistance in the different phases of nitric acid. The corrosion rate of SiN and 304L stainless steels in the liquid phase of nitric acid had a cubic function relationship with temperature, and SiN stainless steel presented better corrosion resistance without intergranular corrosion (IGC) compared with 304L stainless steel with IGC at 100 °C and 120 °C. By contrast, the SiN stainless steel displayed a lower corrosion resistance than 304L stainless steel in the vapor phase of nitric acid at the same temperature, and the corrosion rates of SiN and 304L stainless steels showed a quadratic function relationship with temperature, indicating a milder corrosion in the vapor phase in comparison with that in the liquid phase of nitric acid. In the condensate phase of nitric acid, there was a similar corrosion behavior of the two steels to that in the nitric acid vapor phase, and 304L stainless steel also demonstrated a better corrosion resistance than SiN stainless steel at the same temperature. The differences in corrosion behavior between the two steels could be attributed to the changed media environment and the different alloy composition, and the two aspects were discussed in detail based on relevant experimental results. This work can provide an important insight into the material selection for reprocessing equipment and the development of new corrosion-resistant materials used in spent fuel reprocessing.

**Keywords:** stainless steel; three-phase corrosion behavior; nitric acid; weight loss; corrosion kinetics

## 1. Introduction

Spent fuel reprocessing is the key to achieving nuclear fuel recycling. At present, the mainstream reprocessing method in the world is PUREX or an improved PUREX process. In this method, nitric acid at a high temperature is used to dissolve spent fuel, and tributyl phosphate is adopted to extract uranium and plutonium to remanufacture the new nuclear fuel assemblies after separation and purification [1–3]. Accordingly, the process leaves the spent fuel reprocessing equipment in a service environment of high temperature, strong acidity, strong oxidation and high radioactivity, which requires the materials used for the reprocessing equipment to have superior corrosion resistance [1,2,4,5].

The commonly used materials for spent fuel reprocessing equipment include stainless steels, titanium alloys and zirconium alloys, among which austenitic stainless steel is the most widely used equipment material due to its good mechanical property and low cost [6–12]. However, the intergranular corrosion (IGC) of stainless steels is a common

problem in the spent fuel reprocessing environment because of the existence of nitric acid at high temperature; the native oxidizing ions from spent fuel such as $Ru^{n+}$, $Pu^{4+}$ and $Ce^{4+}$; and the secondary oxidizing ions from stainless steel consisting of $Cr^{6+}$ and $V^{5+}$ [13–15]. Furthermore, with the change in temperature, the IGC effect may be more pronounced [6,13–18]. On the one hand, with the increase in temperature, the reduction reactions of nitric acid and oxidizing ions are accelerated, and the corrosion potential of stainless steel will shift to the anodic direction, resulting in a significant increase in IGC. On the other hand, as the nitric acid temperature rises, three-phase (liquid phase, vapor phase and condensate phase) corrosion occurs, and the flow of nitric acid vapor and the renewal rate of condensate will vary with temperature, causing significant differences in the corrosion rate of materials. Shankar et al. found that the corrosion rate of titanium and titanium alloys in the nitric acid condensate phase was higher than that in the liquid phase and vapor phase [15,19–21]. The main cause was that the periodic renewal of nitric acid condensate led to the low content of $Ti^{3+}$ in the solution, causing difficulty in the formation of a protective passive film and thereby the accelerating corrosion rate. For stainless steels, there may be some differences in the nitric acid three-phase situation. The $Cr^{3+}$ produced by the dissolution of stainless steel will be oxidized to $Cr^{6+}$ in nitric acid at a high temperature, and since $Cr^{6+}$ is an oxidizing ion, it will promote the corrosion of stainless steel [6,22,23]. However, the valence state of $Ti^{3+}$ will not change, and it can participate in the formation of a passive film and inhibit the corrosion of titanium alloys. Therefore, the three-phase corrosion behavior of stainless steel in high-temperature nitric acid should be different from that of titanium alloys, and it needs to be further studied.

In order to improve the IGC resistance of stainless steel in high-temperature nitric acid, it is usually necessary to carry out ultrapure smelting, especially to reduce the content of impurity elements such as C, S and P [24]. This strategy can indeed reduce the tendency of IGC of stainless steel to a certain extent, but it cannot fundamentally solve the problem of IGC of stainless steel in spent fuel reprocessing environment. Recently, a special stainless steel containing about 4 wt.% Si named SiN has been proved to have superior resistance to IGC [5,24]. Laurant et al. found that the SiN stainless steel showed a high resistance to IGC in high-temperature nitric acid containing oxidizing ions [25–28]. Meanwhile, even if the corrosion potential of SiN stainless steel was in the transpassivation zone, IGC still did not occur. Further studies showed that the element Si was evenly distributed in SiN stainless steel, which can hinder the aggregation of carbon and carbide-forming elements such as Cr, Mo and Mn, thus preventing composition inhomogeneity near the grain boundary, especially the chromium-depleted zone [6,25,29]. Moreover, the passive film of the SiN stainless steel contained a certain amount of silicate with a high chemical stability, which made the stainless steel have good corrosion resistance in high-temperature nitric acid [30,31]. However, to the best of the authors' knowledge, the research on the corrosion of SiN stainless steel in high-temperature nitric acid is only limited to the liquid-phase environment. There is still a lack of systematic research on the corrosion behavior of SiN stainless steel in nitric acid vapor and nitric acid condensate, as well as the three-phase corrosion kinetics of SiN stainless steel with the change in temperature.

In this study, we investigated the three-phase corrosion behavior of SiN and 304L stainless steels using a self-built three-phase corrosion apparatus. Nitric acid with oxidizing ions was used to simulate the actual service environment. The corrosion behavior of the tested materials in nitric acid containing oxidizing ions at different temperatures was investigated, and the three-phase corrosion kinetics of the two kinds of stainless steel were revealed. This work can provide an important insight into the material selection for reprocessing equipment and the development of new corrosion-resistant materials used in spent fuel reprocessing.

## 2. Experimental Section

The alloy composition of the solution-annealed SiN and 304L stainless steels used in this experiment is shown in Table 1. The weight loss test was carried out using the hanging

piece method. After grinding to 2000 # using silicon carbide sandpapers and polishing with water-soluble diamond paste with the 1.5 μm particle diameter, the samples were ultrasonically cleaned with anhydrous alcohol, then dried with cold air, and finally weighed. The size of the experimental samples was 10 mm × 10 mm × 3 mm with an exposed area of 3.2 cm$^2$, a hole with a diameter of 0.5 mm was drilled at the edge of the sample, and a string made of polytetrafluoroethylene passed through the small hole to hang the sample. A self-developed nitric acid three-phase corrosion apparatus was adopted for the three-phase corrosion tests, as shown in Figure 1. The device mainly consisted of a heating system, liquid-phase test system, vapor-phase test system and condensate-phase test system. The test solution was first added to a glass-made reaction tank, and then the solution in the liquid-phase test system was heated to different temperatures by the heating system. Some samples were suspended in the liquid-phase test system with the polytetrafluoroethylene braided strings for liquid-phase corrosion tests, and the vapor generated from the solution combined with the NO$_x$ gas generated from the reductive decomposition of nitric acid would corrode the samples housed in the vapor-phase test system. Meanwhile, the nitric acid vapor and NO$_x$ gas would enter the condensation system and be condensed, the condensate would be collected in the condensate-phase test system and the samples housed in it would be corroded. In this way, the corrosion behavior of the samples in each phase could be measured separately. Weight loss tests with three parallel samples were conducted for the three-phase corrosion tests at 20 °C, 60 °C, 80 °C, 100 °C and 120 °C, and the corrosion time was 120 h. After the three-phase corrosion measurements, the corroded samples were washed with deionized water, ultrasonically cleaned in absolute alcohol and weighed after blow-drying to calculate the average corrosion rate of the samples in each phase.

**Table 1.** Composition (wt.%) of SiN and 304L stainless steels used in this work.

| Chemical Composition | C | Cr | Ni | Mo | Mn | Si | S | P | Nb | N | Fe |
|---|---|---|---|---|---|---|---|---|---|---|---|
| SiN | ≤0.010 | 17.74 | 14.88 | 0.30 | 1.39 | 3.84 | ≤0.005 | ≤0.004 | 0.05 | 0.042 | Bal. |
| 304L | ≤0.03 | 18.33 | 10.12 | - | 1.640 | 0.064 | ≤0.004 | ≤0.030 | - | - | Bal. |

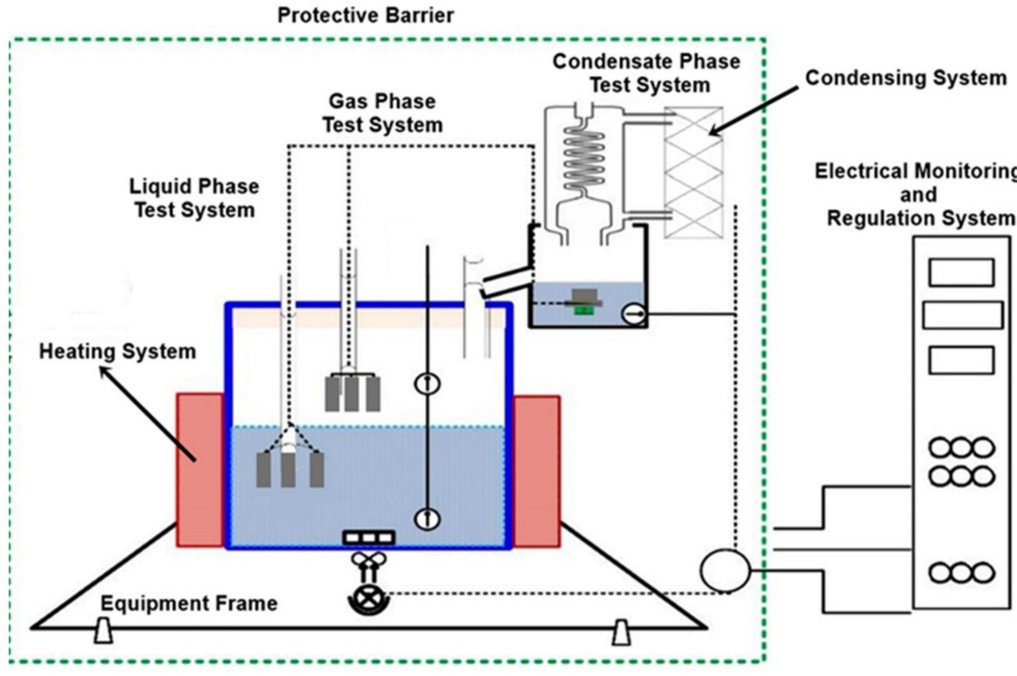

**Figure 1.** High-temperature nitric acid three-phase corrosion apparatus.

Native oxidizing ions with radioactivity such as $Pu^{4+}$ and $Am^{3+}$ were mainly replaced by $V^{5+}$ and $Ce^{4+}$, respectively, because of their similar redox potential, and the amount of ion substitution was calculated based on the number of electron transfers in the redox reaction [7,25,30,32,33]. In order to avoid additional components that can interfere with the experimental system, $V^{5+}$ was added using metal vanadium and $Ce^{4+}$ was added using ceric ammonium nitrate. The secondary oxidizing ion of $Cr^{6+}$ from stainless steel was added using $CrO_3$. Based on the above considerations, the corrosion solution used in this experiment was a 6 mol/L nitric acid solution containing oxidizing ions, and the specific composition of the solution was as shown in Table 2. Additionally, the corrosion morphologies of samples after immersion in the liquid phase, vapor phase and condensate phase of nitric acid for 120 h were observed using a field emission scanning electron microscope (FEI, Waltham, MA, USA), and the surface composition of the samples was analyzed by EDS (FEI, Waltham, MA, USA).

**Table 2.** Composition of the testing solution used in this work.

| Composition | $HNO_3$ | $Cr^{6+}$ | $V^{5+}$ | $Ce^{4+}$ |
|---|---|---|---|---|
| Concentration | 6 mol/L | 0.125 g/L | 1.7 g/L | 1.06 g/L |

## 3. Results and Discussion

### 3.1. Liquid-Phase Corrosion Behavior of Stainless Steels at Different Temperatures

The corrosion rates of the two stainless steels in the liquid phase containing oxidizing ions at different temperatures are shown in Figure 2. It can be found that the corrosion rate of each stainless steel shows an increasing trend with the increase in temperature. Specifically, the corrosion rates of both stainless steels are low with little weight loss at temperatures below 60 °C, while the corrosion rates of both steels increase rapidly from 60 °C to 120 °C. Meanwhile, the corrosion rate of 304L stainless steel is significantly higher than that of SiN stainless steel at the same temperature; the corrosion rate of the former is about 1.34 mm/year (the thickness reduction of materials in each year, and 1 mm/year = 0.039 ipy) at 120 °C, while the latter is about 0.44 mm/year in the same condition. On fitting the corrosion rate data of the two steels at different temperatures, the corrosion rates of SiN and 304L stainless steels both exhibit a cubic function relationship with temperature, as shown in Figure 3, where $y$ represents the corrosion rate; $x$ is the solution temperature; and $a$, $b$, $c$ and $d$ represent the correlation coefficients. Thus, it can be inferred from the function that the corrosion rates of both stainless steels will increase with the elevated solution temperature, and the corrosion rate in nitric acid liquid phase containing oxidizing ions is sensitive to the high-temperature area but less sensitive to the low-temperature area.

Figure 4 shows the corrosion morphologies of SiN stainless steel in a nitric acid solution containing oxidizing ions at different temperatures. It can be seen that the corrosion morphology of SiN steel changes significantly with the change in solution temperature. At 20 °C and 60 °C, the corrosion on the surface is mild, and the grinding scratches are still visible. As the temperature rises to 80 °C, the surface corrosion is greatly accelerated, and the entire surface is covered in white corrosion products. However, the surface corrosion products of SiN steel are significantly reduced when the temperature increases to 100 °C, a small number of corrosion spots appear and some grain boundaries become obvious. By combining these results with the changed curve of corrosion rate with temperature in Figure 3, it can be inferred that the reduced corrosion product can be attributed to the rapid surface dissolution of SiN steel with the rising solution temperature, displaying some small corrosion spots and clear grain boundaries. When the temperature increases to 120 °C, larger corrosion spots and obvious grain boundaries are observed, indicating more severe corrosion. Based on the results of morphology, the corrosion of SiN stainless steel becomes more severe with increased temperature, which is consistent with the result of corrosion rate. It should be noted that although some grain boundaries are observed at 100 °C and

120 °C, no significant IGC occurs on the surface of SiN stainless steel, which is in line with the results of Laurant et al. [25,28]. In addition, the surface composition of SiN stainless steel was further analyzed by EDS (Figure 5), and the results show that the elements Cr, Fe and Ni exist, implying that the surface corrosion products correspond to the oxides of Cr, Fe and Ni.

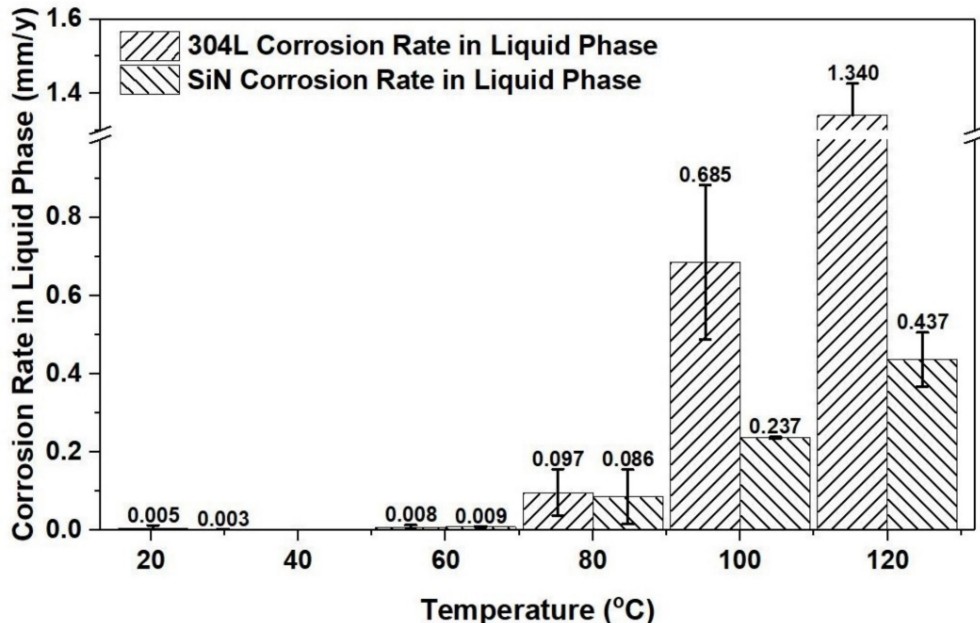

**Figure 2.** Corrosion rates of SiN and 304L stainless steels in nitric acid liquid phase at different temperatures.

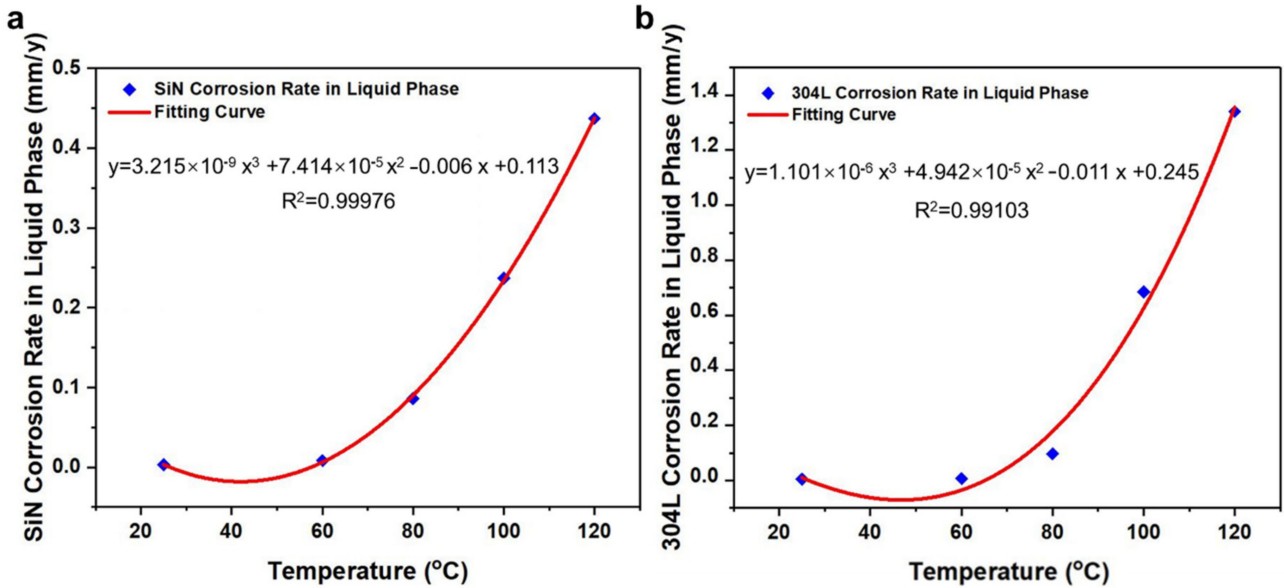

**Figure 3.** Corrosion rate fitting curves of SiN and 304L stainless steels in nitric acid liquid phase at different temperatures, (**a**) SiN stainless steel and (**b**) 304L stainless steel.

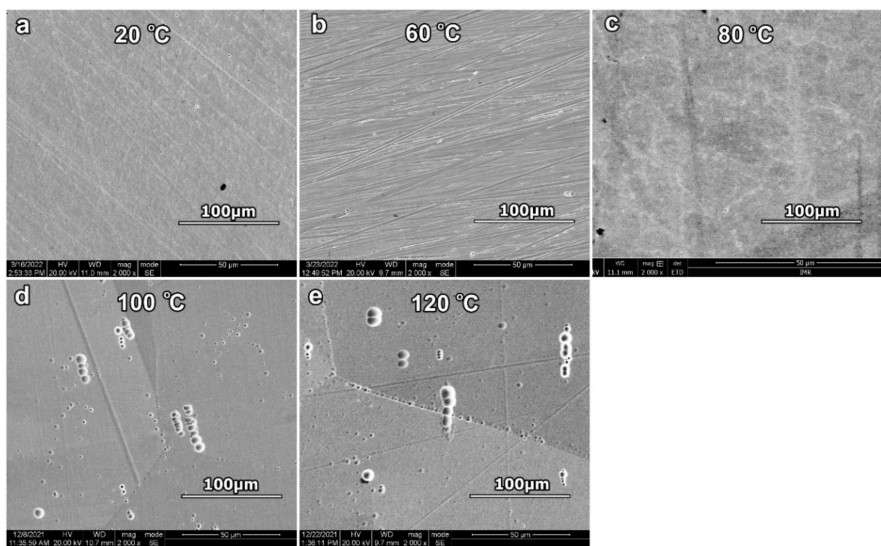

**Figure 4.** Corrosion morphologies of SiN stainless steel after immersion in nitric acid liquid phase at different temperatures for 120 h, (**a**) 20 °C, (**b**) 60 °C, (**c**) 80 °C, (**d**) 100 °C and (**e**) 120 °C.

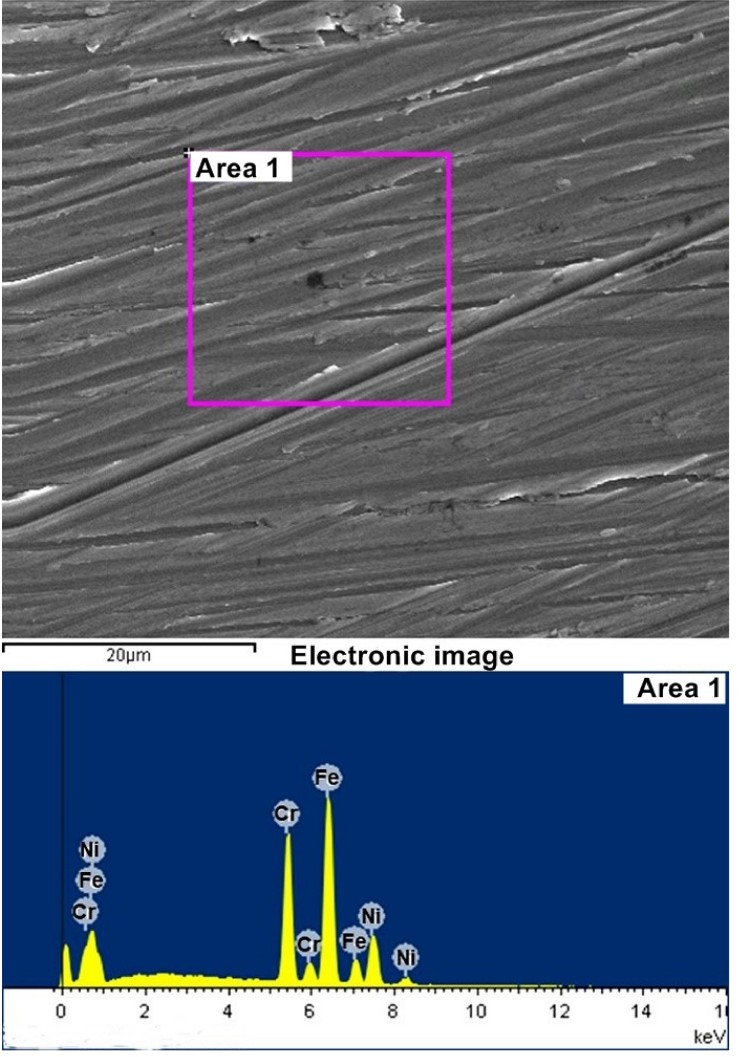

**Figure 5.** Composition of surface film of SiN stainless steel in nitric acid liquid phase.

The corrosion morphologies of 304L stainless steel in the nitric acid liquid phase containing oxidizing ions at different temperatures are shown in Figure 6. It is clear that the corrosion morphology of 304L stainless steel exhibits a significant difference with the change in solution temperature in the nitric acid liquid phase. When the temperature is lower than 60 °C, only slight uniform corrosion occurs on the surface of 304L stainless steel, and the whole sample surface is smooth without obvious corrosion products. When the temperature rises to 80 °C, obvious corrosion products can be observed on the surface of 304L stainless steel, which is similar to the morphology of SiN stainless steel. As the temperature increases to 100 °C, 304L stainless steel shows obvious IGC leaving deeper corrosion grooves along the grain boundaries, and some corrosion spots in the grain interior are also observed. With the temperature increasing to 120 °C, 304L stainless steel exhibits a similar corrosion morphology to that of 100 °C, while the IGC is more serious with wider corrosion grooves at grain boundaries, and the corrosion spots inside the grains become larger. The results of EDS (Figure 7) reveal the elements Cr, Ni, Mn and Fe, showing a slight difference from the composition of SiN stainless steel. Specifically, the element Mn was not detected on the surface of SiN stainless steel. The main role of Mn in stainless steel is to enlarge the austenite zone and improve its quenching permeability. Meanwhile, Mn can promote carbide formation, and the existence of carbide is detrimental to stainless steel IGC resistance, which can explain the rationale for IGC in 304L stainless steel. In summary, the results of corrosion rate and corrosion morphology show that the corrosion resistance of SiN stainless steel is significantly better than that of 304L stainless steel in the nitric acid liquid phase above 80 °C.

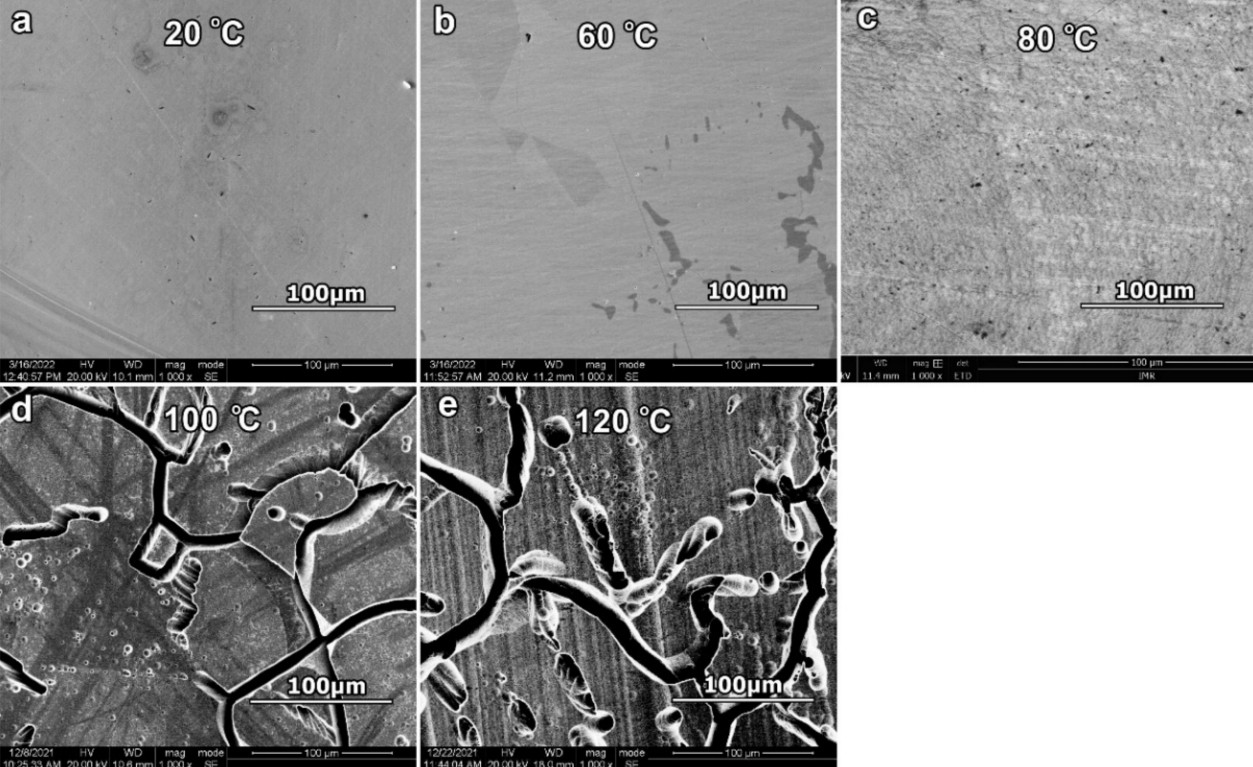

**Figure 6.** Corrosion morphologies of 304L stainless steel after immersion in nitric acid liquid phase at different temperatures for 120 h, (**a**) 20 °C, (**b**) 60 °C, (**c**) 80 °C, (**d**) 100 °C and (**e**) 120 °C.

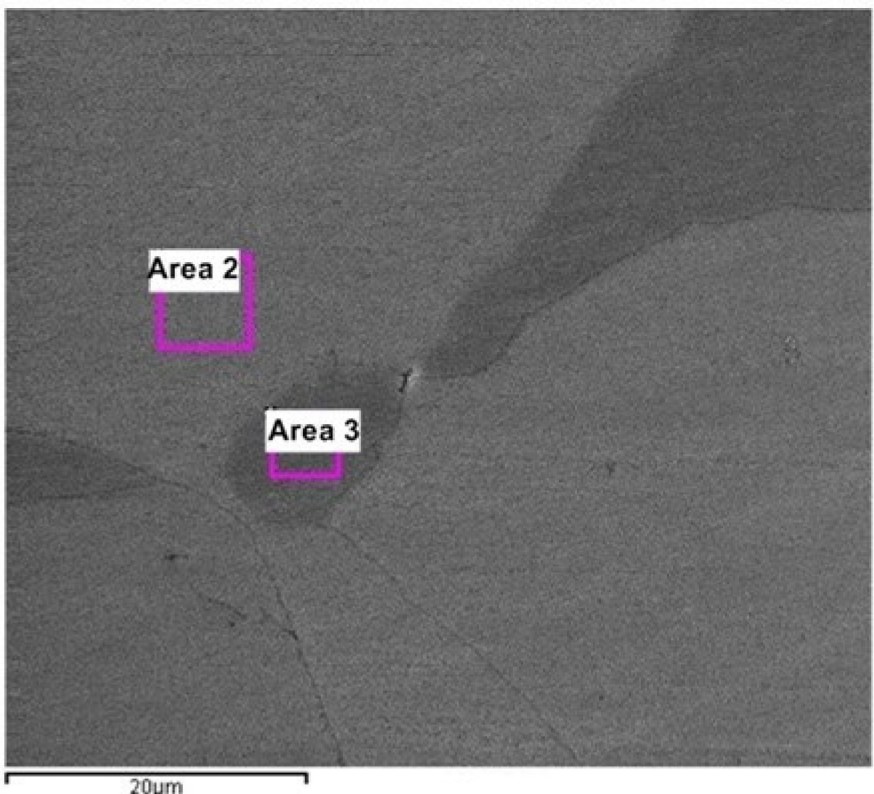

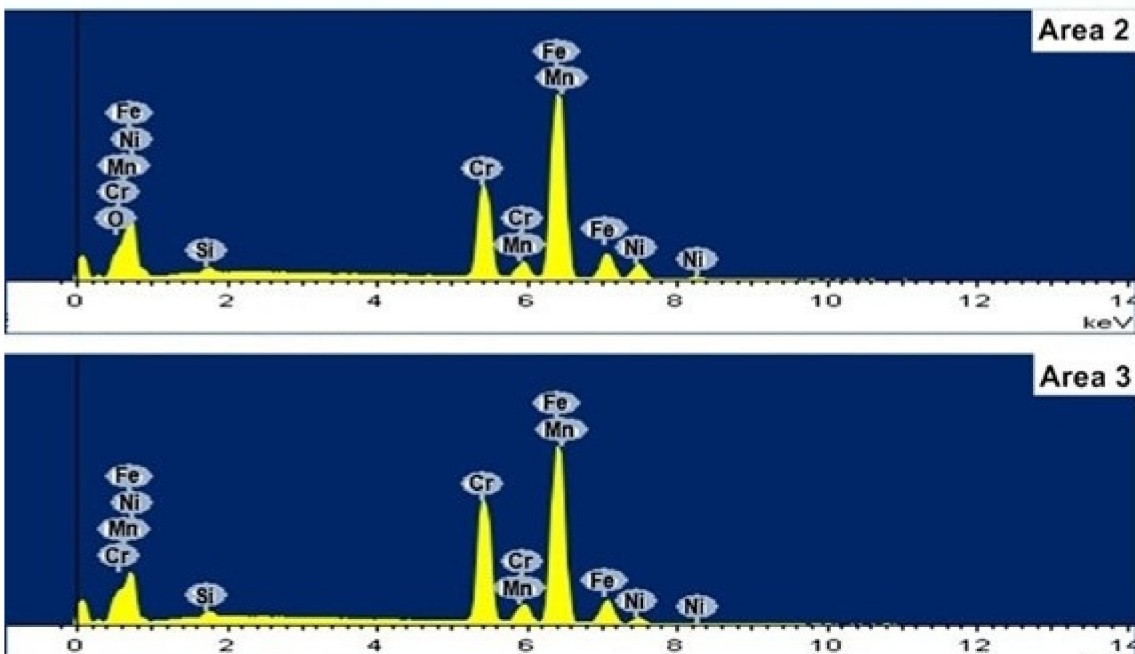

**Figure 7.** Composition of surface film of 304L stainless steel in nitric acid liquid phase.

### 3.2. Vapor-Phase Corrosion Behavior of Stainless Steels at Different Temperatures

The corrosion behavior of SiN and 304L stainless steels in the vapor phase of nitric acid was also investigated. It is known that the nitric acid vapor will increase with the elevated temperature, and thus only four temperatures, namely 60 °C, 80 °C, 100 °C and 120 °C, were selected for the vapor-phase corrosion study considering the extremely low content of nitric acid vapor at room temperature. The corrosion rates of SiN and 304L stainless steels in the vapor phase of nitric acid at different temperatures are shown in

Figure 8, and both steels show a significant increase with the increasing temperature. Interestingly, the corrosion rate of SiN stainless steel is obviously higher than that of 304L stainless steel at the same temperature, which is contrary to the results in the nitric acid liquid phase. At 100 °C and 120 °C, the SiN stainless steel has the corrosion rates of 0.01 mm/year and 0.126 mm/year, respectively, which are approximately twice as high as those of 304L stainless steel corresponding to 0.005 mm/year and 0.054 mm/year, respectively, at the same temperature. Meanwhile, all the corrosion rates in the nitric acid vapor phase of the two steels are smaller than those in the nitric acid liquid phase at the same temperature, which is mainly attributed to the difference in environment between the two phases. For the nitric acid liquid phase, the presence of a large number of oxidizing ions can significantly accelerate the corrosion of stainless steel, especially the IGC. However, there are no oxidizing ions in the vapor phase of nitric acid, so the corrosion rate in the vapor phase is relatively low. On the other hand, the reason for SiN stainless steel corroding at a higher rate in the nitric acid vapor phase than 304L stainless steel but showing the opposite result in the liquid phase of nitric acid could be related to the alloy composition difference between the two steels. For SiN stainless steel, about 4 wt.% Si was added into the steel, and the addition of Si mainly inhibits the IGC caused by oxidizing ions. However, Si is not a corrosion-resistant element, and its addition will increase the uniform corrosion rate of SiN steel to a certain degree [16,34,35]. Thus, the corrosion rate of SiN stainless steel in the nitric acid liquid phase containing oxidizing ions is lower than that of 304L stainless steel, while it is higher than that of 304L stainless steel in the nitric acid vapor phase without oxidizing ions.

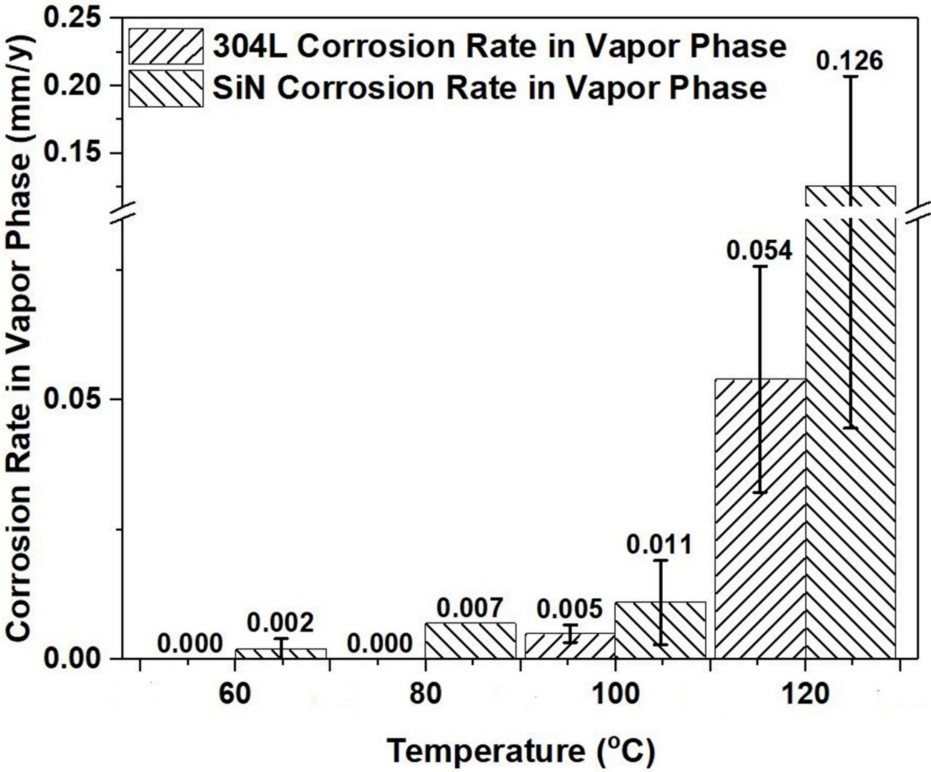

**Figure 8.** Corrosion rates of SiN and 304L stainless steel in nitric acid vapor phase at different temperatures.

Furthermore, when the corrosion rates of the two stainless steels at different temperatures are fitted, a quadratic function relationship between the corrosion rate and the temperature for the two stainless steels is displayed, as shown in Figure 9, where $y$ represents the corrosion rate; $x$ is the temperature; and $a$, $b$, $c$ and $d$ represent the correlation coefficients. Based on the fitting results, it can be inferred that the corrosion rate of the two stainless steels will increase with the increase in temperature. However, it needs to

be stated that the corrosion rate of the two stainless steels is lower in the nitric acid vapor phase compared to the cubic function relationship in the nitric acid liquid phase, which also illustrates to some extent that the corrosion in the nitric acid liquid phase is higher than that in the nitric acid vapor phase.

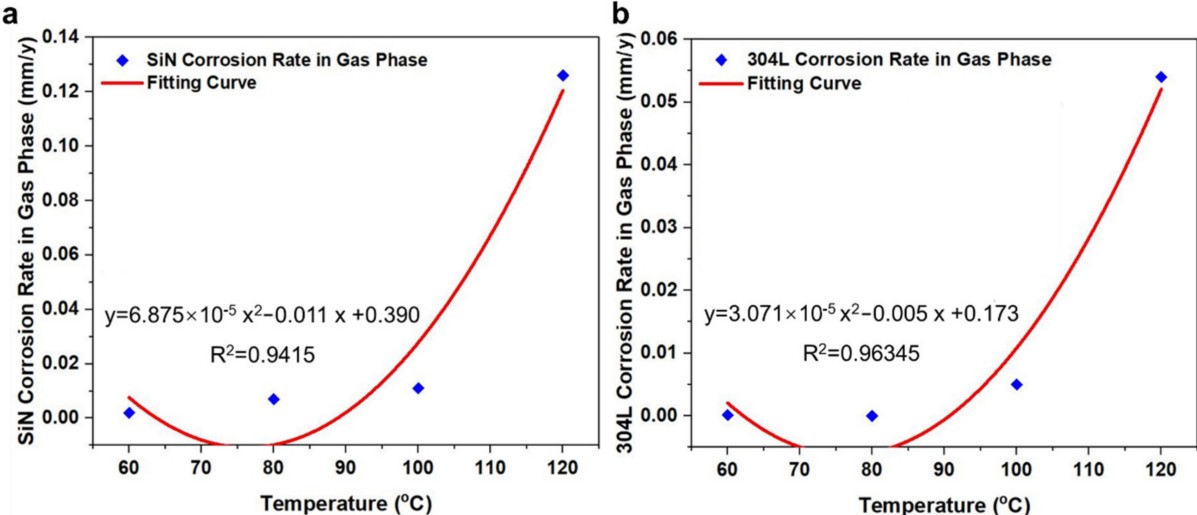

**Figure 9.** Corrosion rate fitting curves of SiN and 304L stainless steels in nitric acid vapor phase at different temperatures, (**a**) SiN stainless steel and (**b**) 304L stainless steel.

Figure 10 presents the corrosion morphologies of SiN stainless steel corroded for 120 h in the nitric acid vapor phase at different temperatures. As shown, only slight corrosion occurs on the surface of SiN steel at 60 °C, and obvious scratches from grinding in the sample preparation process can be observed. The corrosion products remarkably increase when the temperature rises to 80 °C, but the scratches on the sample surface can still be seen. As the temperature increases to 100 °C, the corrosion of the sample becomes more severe; however, no IGC is observed on the sample surface. Clear grain boundaries and a few small corrosion spots are observed on the surface when the temperature increases to 120 °C. Figure 11 presents the corrosion morphologies of 304L stainless steel corroded for 120 h in the nitric acid vapor phase at different temperatures. From the corrosion morphology, it is apparent that the corrosion degree of 304L stainless steel increases gradually with the rising temperature; however, there is no sign of IGC even at the highest temperature of 120 °C, indicating that 304L stainless steel exhibits superior corrosion resistance in the nitric acid vapor phase. Studies have shown that the principal problem for the application of 304L stainless steel in high-temperature nitric acid is IGC, which will be more serious in the nitric acid solution containing oxidizing ions. Different from the liquid-phase environment, there are no oxidizing ions in the vapor phase, so 304L stainless steel still maintains superior corrosion resistance in the nitric acid vapor phase without IGC. This result can provide an important guide for the application of 304L stainless steel in a spent fuel reprocessing vapor-phase environment. In summary, combining the results of corrosion rate and corrosion morphology, it can be concluded that 304L stainless steel exhibits better corrosion resistance than SiN stainless steel at the same temperature in the nitric acid vapor-phase environment.

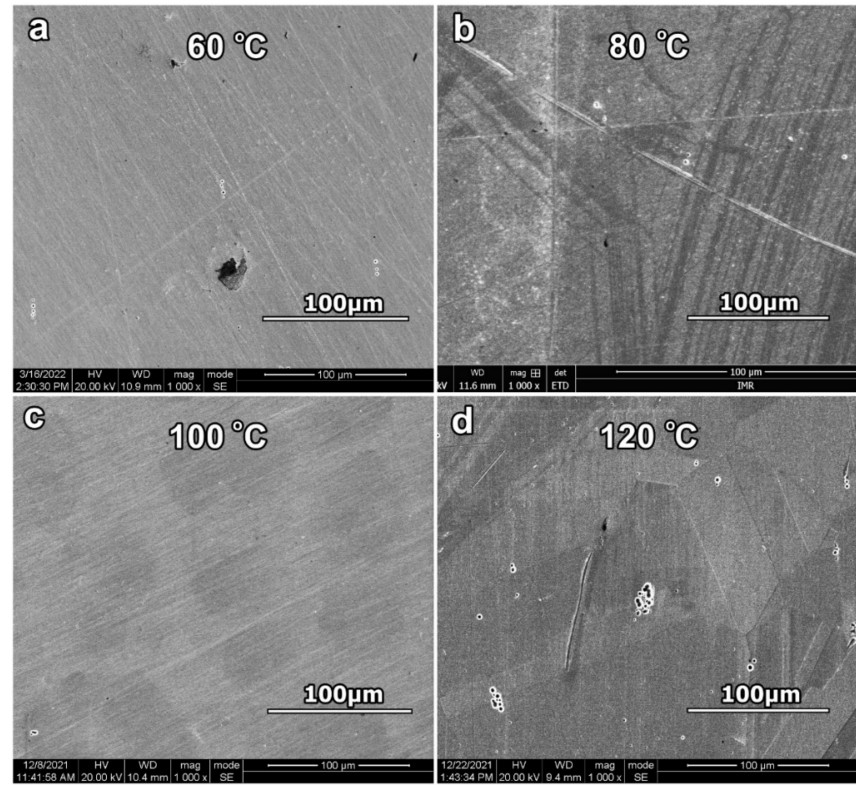

**Figure 10.** Morphologies of SiN stainless steel after corrosion in nitric acid vapor phase at different temperatures for 120 h, (**a**) 60 °C, (**b**) 80 °C, (**c**) 100 °C and (**d**) 120 °C.

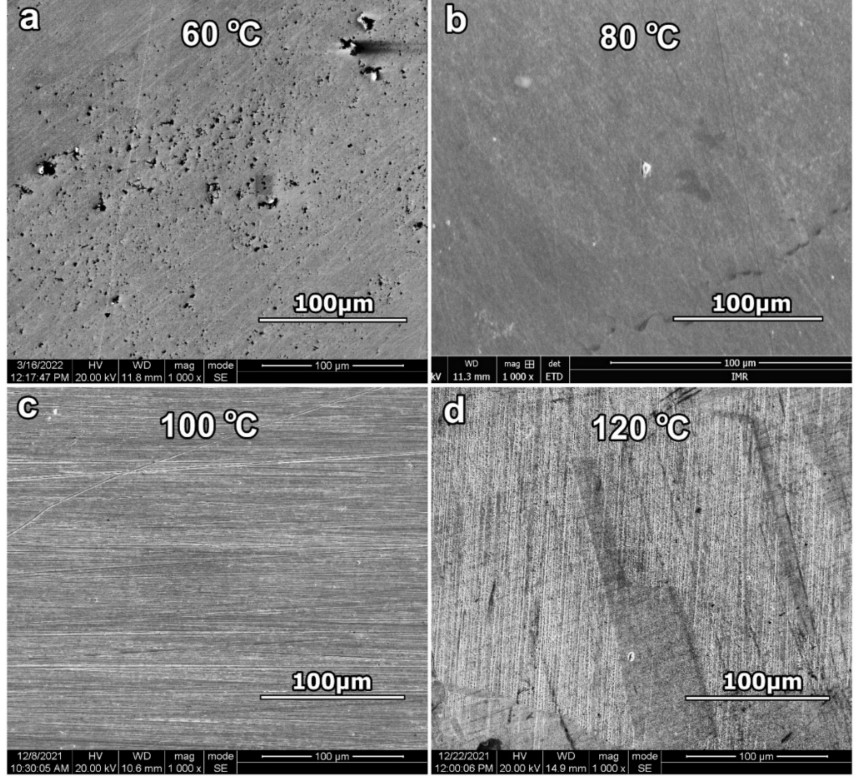

**Figure 11.** Morphologies of 304L stainless steel after corrosion in nitric acid vapor phase at different temperatures for 120 h, (**a**) 60 °C, (**b**) 80 °C, (**c**) 100 °C and (**d**) 120 °C.

### 3.3. Condensate-Phase Corrosion Behavior of Stainless Steels at Different Temperatures

The composition of the condensate phase is significantly affected by the experimental temperature. As the temperature increases, the concentration of nitric acid in the condensate phase increases and the renewal rate of condensate becomes faster. In view of the minimum amount of condensate collected at room temperature, four temperatures, namely 60 °C, 80 °C, 100 °C and 120 °C, were selected for the condensate-phase corrosion tests. The corrosion rates of SiN and 304L stainless steels in the condensate phase at different temperatures are shown in Figure 12. From the results obtained, it is obvious that the corrosion rates of both stainless steels show an increasing trend with the increase in temperature. The corrosion rate of SiN stainless steel approaches zero below 80 °C, while it increases rapidly with the increasing temperature. The corrosion rates of SiN stainless steel are about 0.023 mm/year and 0.078 mm/year at 100 °C and 120 °C, respectively. With regard to 304L stainless steel, it displays superior corrosion resistance with a corrosion rate close to zero at 60 °C, and the corrosion rate increases significantly with the increasing temperature. Notably, with the temperature increasing to 100 °C and 120 °C, the corrosion rate of 304L stainless steel in the nitric acid condensate phase reaches 0.010 mm/year and 0.035 mm/year, respectively, which is significantly lower than that of SiN stainless steel at the same temperature. The reason for the difference in the corrosion rate between the two stainless steels may be closely related to the changed composition of the nitric acid condensate phase compared with that of the nitric acid liquid phase. The concentration of nitric acid in the condensate phase will increase with the elevated temperature. Meanwhile, the nitric acid condensate phase usually has fewer oxidizing ions compared with the nitric acid liquid phase, and thus the two steels will suffer from uniform corrosion in the condensate phase [14,36]. However, because SiN stainless steel contains about 4 wt.% Si, it will corrode faster at higher temperatures. The corrosion rates of SiN and 304L stainless steels in the nitric acid condensate phase at different temperatures are fitted as shown in Figure 13. It can be seen that the corrosion rates of both stainless steels in the nitric acid condensate phase exhibit a quadratic function with temperature, which is similar to the case in the nitric acid vapor phase. Based on the corrosion rate, the 304L stainless steel presents better corrosion resistance in the condensation phase, especially when the temperature is higher than 80 °C.

Figure 14 presents the corrosion morphologies of SiN stainless steel in the nitric acid condensate phase at different temperatures. It can be seen that the corrosion morphology of SiN stainless steel exhibits some differences with the increasing temperature. The whole surface of SiN stainless steel displays a uniform corrosion morphology without corrosion spots below 100 °C. Minor corrosion spots are observed at 100 °C, and as the temperature rises to 120 °C, the number of corrosion spots on the surface increases, and some grain boundaries appear. The corrosion morphologies of 304L stainless steel in nitric acid condensate phase at 60 °C, 80 °C, 100 °C and 120 °C are shown in Figure 15. All morphologies show uniform corrosion in the condensate phase. Some corrosion products can be seen on the surface at 100 °C, and the corrosion products disappear at a higher temperature of 120 °C. This can be attributed to the increased nitric acid concentration in the condensate phase because of the rising experimental temperature, which promotes the dissolution of the corrosion products formed at a low temperature. For this section, based on the results of corrosion rate and corrosion morphology, it is clear that 304L stainless steel exhibits better corrosion resistance than SiN stainless steel at the same temperature in the nitric acid condensate phase.

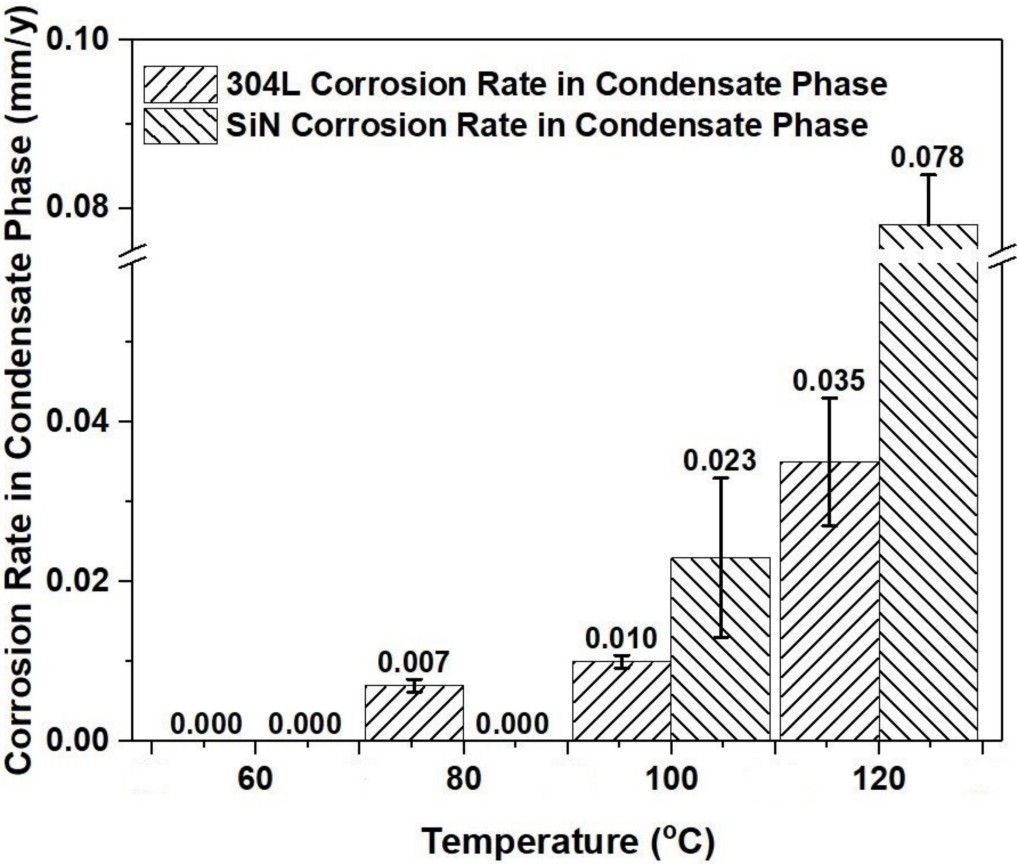

**Figure 12.** Corrosion rates of SiN and 304L stainless steels in nitric acid condensate phase at different temperatures.

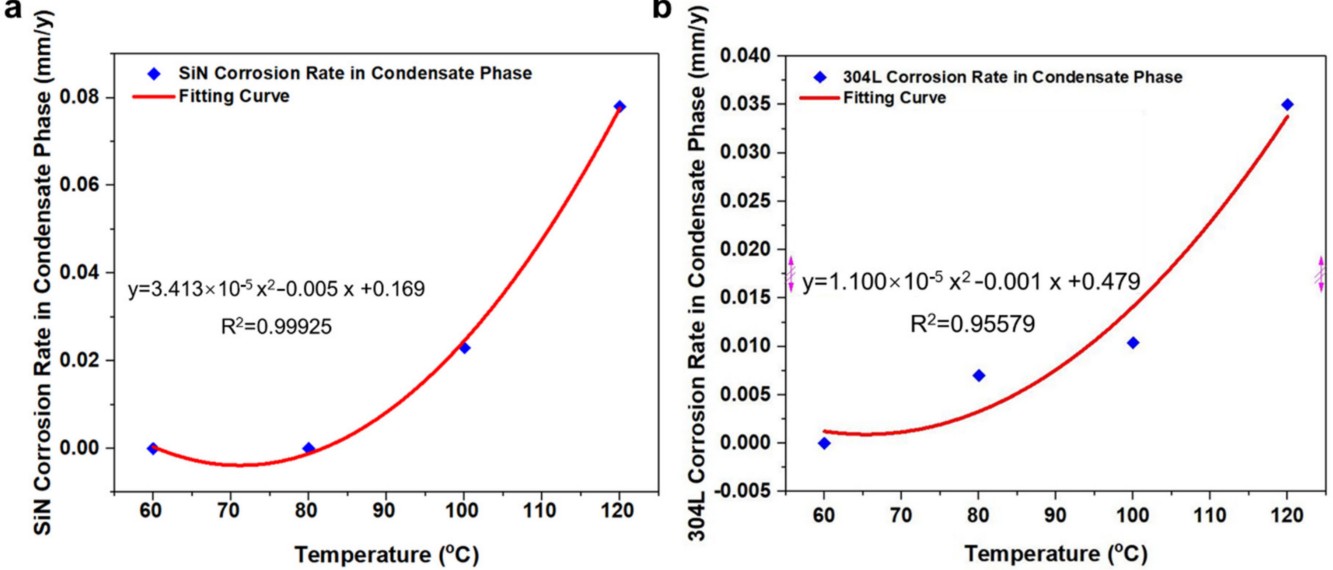

**Figure 13.** Corrosion rate fitting curves of SiN and 304L stainless steels in nitric acid condensate phase at different temperatures, (**a**) SiN stainless steel and (**b**) 304L stainless steel.

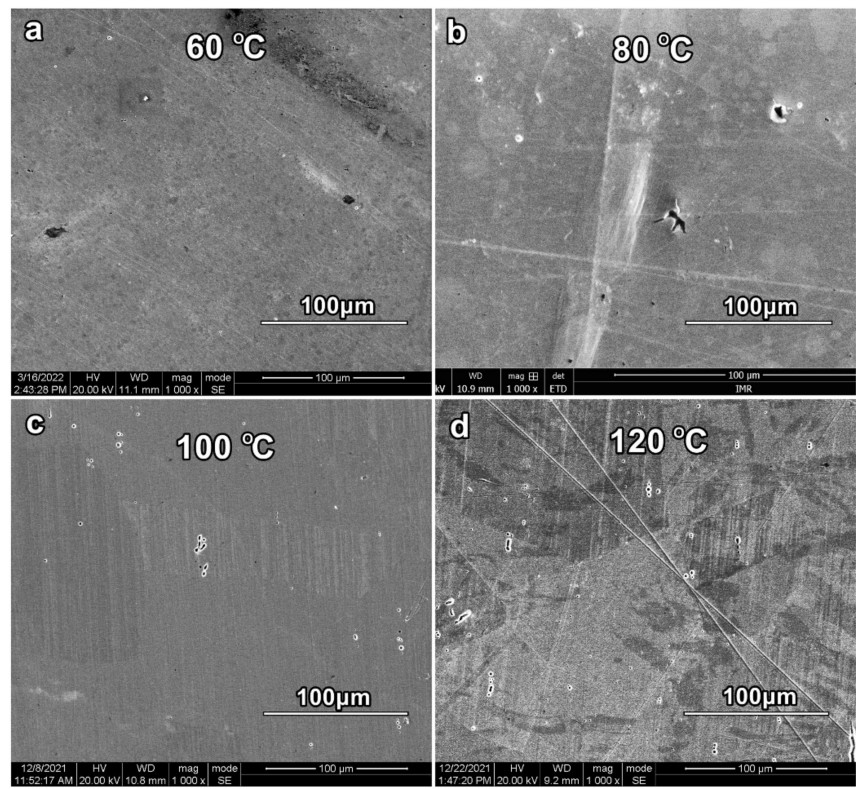

**Figure 14.** Morphology of SiN stainless steel corroded in condensate phase at different temperatures for 120 h, (**a**) 60 °C, (**b**) 80 °C, (**c**) 100 °C and (**d**) 120 °C.

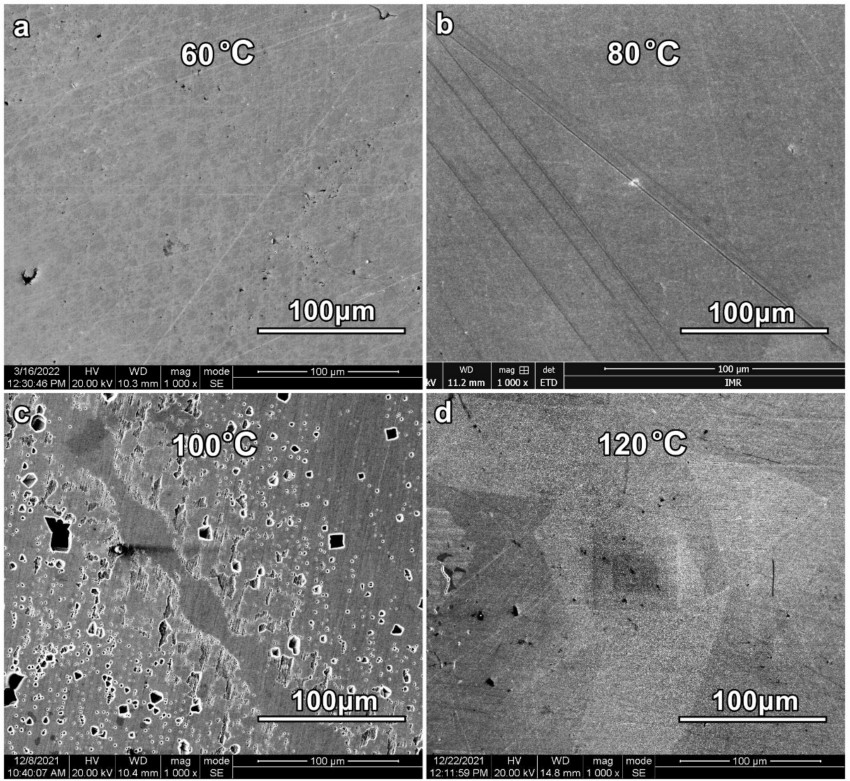

**Figure 15.** Morphology of 304L stainless steel corroded in condensate phase at different temperatures for 120 h, (**a**) 60 °C, (**b**) 80 °C, (**c**) 100 °C and (**d**) 120 °C.

## 4. Conclusions

This work reported the three-phase corrosion behavior of SiN and 304L stainless steels in a nitric acid environment at different temperatures, and the variation in three-phase corrosion kinetics of these two stainless steels with temperature was revealed. The main findings and conclusions are as follows:

(1)  The corrosion rate of SiN stainless steel in the nitric acid liquid phase containing oxidizing ions was significantly lower than that of 304L stainless steel at the same temperature, and SiN stainless steel did not undergo IGC at 100 °C and 120 °C, whereas 304L stainless steel did undergo IGC at the same temperature. The corrosion rate of SiN and 304L stainless steels in the nitric acid liquid phase had a cubic function relationship with temperature, indicating that the corrosion rates of both stainless steels in the liquid phase were sensitive to the high-temperature area but less sensitive to the low-temperature area.

(2)  The corrosion rate of SiN stainless steel in the nitric acid vapor phase was obviously higher than that of 304L stainless steel at high temperatures. The corrosion rates of SiN stainless steel at 100 °C and 120 °C were about twice that of 304L stainless steel. The reason should be correlated with the alloy composition difference between the two steels. The addition of about 4 wt.% Si to SiN stainless steel would increase its uniform corrosion rate to some extent. In addition, the corrosion rates of SiN and 304L stainless steels in the nitric acid vapor phase showed a quadratic function relationship with temperature, which was different from the cubic function relationship in the nitric acid liquid phase containing oxidizing ions, implying that the degree of corrosion of both stainless steels in the nitric acid vapor phase was milder than that in the nitric acid liquid phase.

(3)  The corrosion rates of SiN and 304L stainless steels in the nitric acid condensate phase showed an increasing trend with the rising temperature, with no evidence of IGC. The corrosion rate of SiN stainless steel was higher than that of 304L stainless steel at the same temperature, indicating that 304L stainless steel had a better corrosion resistance in the nitric acid condensate phase. The corrosion rates of both stainless steels in the condensate phase showed a quadratic function relationship with temperature, which was similar to the case in the nitric acid vapor phase.

**Author Contributions:** S.S.: experimental investigation, data analysis. L.Z.: conceptualization, writing—review and editing. A.M.: formal analysis, experimental investigation. E.F.D.: writing—review and editing. C.Z.: conceptualization, supervision. Y.Z.: conceptualization, supervision. All authors have read and agreed to the published version of the manuscript.

**Funding:** This research was funded by China National Nuclear Corporation (CNNC) Science Fund for Talented Young Scholars, the National Natural Science Foundation of China (52101105) and the Chinese Postdoctoral Science Foundation (2020M670811).

**Data Availability Statement:** The raw and the processed data are available from Lianmin Zhang (Email: lmzhang14s@imr.ac.cn) upon reasonable request.

**Acknowledgments:** We thank Yiyin Shan, Wei Yan, Yanfen Li and Jiarong Zhang for providing the experimental materials.

**Conflicts of Interest:** The authors declare no conflict of interest.

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
