# Peer review of "Comparison of the Three-Phase Corrosion Behavior of SiN and 304L Stainless Steels in 6 M Nitric Acid Solution at Different Temperatures"

_metals, doi:10.3390/met12060922_

Round 1

Reviewer 1 Report

  1. Why were these test temperatures chosen?
    2. The formation of what phases/ions can explain the difference in corrosion rates in SiN and 304L steels in the vapor phase?
    3. What are the corrosion products observed in the SEM?                         4. Why was this concentration of the corrosive solution chosen?

Author Response

Dear Reviewer,

Thank you for your letter and comments concerning our manuscript entitled “Comparison of the three-phase corrosion behavior of SiN and 304L stainless steels in 6 M nitric acid at different temperatures(metals-1713076). Those comments are all valuable and very helpful for improving our paper. We have carefully studied the comments and have made appropriate corrections in line with the recommendations of reviewers. Changes are highlighted with yellow background in the revised version. The main corrections in the manuscript and the responses to your comments are as follows:

Reviewer 1

Comment 1: Why were these test temperatures chosen?

Reply: Thank you for your comment. We choose the test temperatures based on the operating conditions of spent fuel reprocessing device, which involve a temperature range from room temperature to 120 oC. For instance, the operating temperatures of evaporator, dissolver and acid-recovery device are 120 oC, 100 oC and room temperature (20 oC), respectively.

Comment 2: The formation of what phases/ions can explain the difference in corrosion rates in SiN and 304L steels in the vapor phase?

Reply: Thanks for your comment. The different corrosion rates between SiN stainless steel and 304L stainless steel can be attributed to the difference in alloy composition. With regard to SiN stainless steel, it contains about 4 wt.% Si, while there is no Si in 304L stainless steel. The main purpose of adding Si element is to inhibit the intergranular corrosion caused by oxidizing ions. However, Si element itself is not a corrosion-resistant element, it can cause an increasing uniform corrosion rate of SiN stainless steel in nitric acid [1-3]. In the vapor phase of nitric acid, the major corrosion type is uniform corrosion because there are no oxidizing ions, so the SiN stainless steel presents a larger corrosion rate compared with that of 304L stainless steel.

[1] Robin, R.; Miserque, F.; Spagnol, V. Correlation between composition of passive layer and corrosion behavior of high Si-containing austenitic stainless steels in nitric acid. J. Nucl. Mater. 2008, 375, 65-71, doi:10.1016/j.jnucmat.2007.10.016

[2] B. Laurent, N. Gruet, B. Gwinner, F. Miserque, V. Soares-Teixeira, K. Ogle, Silicon enrichment of an austenitic stainless steel - impact on electrochemical behavior in concentrated nitric acid with oxidizing ions, Electrochim. Acta 322 (2019) 134703. https://doi.org/10.1016/j.electacta.2019.134703

[3] B. Laurent, N. Gruet, B. Gwinner, F. Miserque, K. Rousseau, K. Ogle, The kinetics of transpassive dissolution chemistry of stainless steels in nitric acid: the impact of Si, Electrochim. Acta 258 (2017) 653-661. https://doi.org/10.1016/j.electacta.2017.11.110

Comment 3: What are the corrosion products observed in the SEM?

Reply: It is a good question. According to the EDS results, there exist obvious element peaks of Fe, Cr, Ni and O on the surface of SiN stainless steel, which means that the corrosion products should be related to the oxides or hydroxide of Fe, Cr and Ni. For 304L stainless steel, there are the element peaks of Fe, Cr, Ni, Mn and O on the surface, indicating that the corrosion products should be correlated with the oxides and hydroxides of Fe, Cr, Ni and Mn. Furthermore, our recent research results of XPS (the following figure) indicate that the main corrosion products of SiN or 304L stainless steel are the oxides and hydroxides of Fe and Cr, which is in line with the result of EDS.

Figure. XPS results of SiN steel after immersion in nitric acid for 48 h at 95 oC.

Comment 4: Why was this concentration of the corrosive solution chosen?

Reply: Thanks for your question. The concentration of corrosive solution used in this work was based on the major operating condition. During the actual operation of the reprocessing unit, 6 mol/L HNO3 is used to dissolve the spent fuel, so we choose the same concentration of HNO3. In addition, the oxidizing ions selected in this work are also based on the actual liquid composition.

It is our hope that our responses to your comments and modifications on the manuscript can be approved by you. Thank you for the opportunity.

Sincerely,

L.M. Zhang

Associate Professor

CAS Key Laboratory of Nuclear Materials and Safety Assessment

Institute of Metal Research, CAS

62 Wencui Road, Shenyang 110016, China

Tel: +86 024-23915904, +86 17609864684

Email: lmzhang14s@imr.ac.cn

Reviewer 2 Report

The manuscript, entitled „ Comparison of the Three-Phase Corrosion Behavior of SiN and 304L Stainless Steels in 6 M Nitric Acid at Different Temperatures” is relevant to the scope of this journal. It is an interesting study that can bring valuable information to specialists

However, some points need to be addressed prior to publication of this manuscript. My comments/suggestions are given:

  1. The authors have made a synthesis of the literature that provides an overview of the research evolution in this area. The authors could enrich the bibliographical references with other existing works in the literature in the field studied.
  2. In Figure 2, the authors should specify in the caption that errors appear in the images. Same for Figure 8.
  3. I don't understand the unit of measurement for corrosion rate "mm/a". Please explain.
  4. If the polynomial dependence of corrosion rate with temperature has been established, why don't you extract from it some important parameters for the kinetics of corrosion? Otherwise, I don't see why these fittings were made!
  5. The evidence of oxide formation on the surface of alloys, as is the case for 304L stainless steel after immersing in nitric acid liquid phase (see Figures 6 and 7) must be supplemented with XRD analysis to be able to specify exactly which oxides have formed.
  6. Why doesn't the error bar appear in Figure 12? Please complete.
  7. The equations from Figures 3, 9 and 12 are not visible. Please correct.
  8. The authors should compare their results with other existing references in the literature.
  9. Why was EDS analysis not performed for each type of corrosion studied? Both EDS and XRD analyses should be completed in the manuscript.
  10. Some of the bibliographic references have errors. I recommend checking them carefully. For example. Ref. 20, 21.

Author Response

Dear Reviewer,

Thank you for your letter and comments concerning our manuscript entitled “Comparison of the three-phase corrosion behavior of SiN and 304L stainless steels in 6 M nitric acid at different temperatures(metals-1713076). Those comments are all valuable and very helpful for improving our paper. We have carefully studied the comments and have made appropriate corrections in line with the recommendations of reviewers. Changes are highlighted with yellow background in the revised version. The main corrections in the manuscript and the responses to your comments are as follows:

Reviewer 2

Comment 1: The authors have made a synthesis of the literature that provides an overview of the research evolution in this area. The authors could enrich the bibliographical references with other existing works in the literature in the field studied.

Reply: Thanks for your comment. We have combed and added relevant references in the latest manuscript.

Comment 2: In Figure 2, the authors should specify in the caption that errors appear in the images. Same for Figure 8.

Reply: Thanks. According to your suggestion, we have modified Figure 2 and Figure 8 in the new version.

Comment 3: I don't understand the unit of measurement for corrosion rate "mm/a". Please explain.

Reply: The unit "mm/a" is one of the commonly used units of corrosion rate [1-3]. In order to make the expression clearer, we also explain it in the text. It means the material thickness lost every year, where "a" refers to "a year". The calculation formula of mm/a is as follows:

X- Corrosion rate (mm/a)

W1-Weight of sample before test (g)

W2-Weight of sample after test (g)

A-Exposed area of sample (cm2)

T-Test duration (h)

D-Material density of test sample (g/cm3)

87600-Calculation constant.

PS: 1 mm/a = 0.039 ipy.

[1] M Takeuchi et al., Corrosion Study of Titanium-5% Tantalum Alloy in Hot Nitric Acid Condensate. In Proceedings of the International Conference on Nuclear Engineering, 2012; pp. 335-340.

[2] AR Shankar et al., Refractory metal coatings on titanium to improve corrosion resistance in nitric acid medium. 2013, 235, 155-164.

[3] Y Sano et al., Effect of metal ions in a heated nitric acid solution on the corrosion behavior of a titanium–5% tantalum alloy in the hot nitric acid condensate. J. Nucl. Mater. 2013, 432, 475-481, doi:10.1016/j.jnucmat.2012.08.009.

Comment 4: If the polynomial dependence of corrosion rate with temperature has been established, why don't you extract from it some important parameters for the kinetics of corrosion? Otherwise, I don't see why these fittings were made!

Reply: Thanks for your comment. The purpose of establishing the polynomial dependence of corrosion rate with temperature is to reveal the change law of corrosion kinetics and to compare the corrosion degree in different phases at the same temperature between SiN stainless steel and 304L stainless steel. For instance, we can obtain the corrosion rate of the two stainless steels at 110 oC based the established fitting equations, although we have not conducted relevant tests. Meanwhile, we can also qualitatively evaluate the influence of temperature on the corrosion degree of stainless steels according to the size of polynomial order. Generally, the greater the polynomial order, the more serious the corrosion.

After considering your comment, we have revised this part and compared the three-phase corrosion degree between the two stainless steels at the same temperature.

Comment 5: The evidence of oxide formation on the surface of alloys, as is the case for 304L stainless steel after immersing in nitric acid liquid phase (see Figures 6 and 7) must be supplemented with XRD analysis to be able to specify exactly which oxides have formed.

Reply: It is a nice advice. We totally agree with you. Indeed, if the XRD analysis can be added in the manuscript, the evidence of oxide formation on the surface of stainless steels can be more convincing. Actually, we tried the XRD tests of the corrosion surface before, while it is difficult to obtain the real composition of oxides on the surface due to the thin thickness of oxide films with several nanometers. Also, the XRD parallel light tests failed because the flatness of the mass loss samples with a large size cannot be effectively controlled. So, we did not provide the results of XRD in this work.

For all that, the results of EDS can reveal the composition of the oxide films of stainless steels to a certain extent. According to the EDS results, there exist obvious element peaks of Fe, Cr, Ni and O on the surface of SiN stainless steel, which means that the corrosion products should be related to the oxides or hydroxide of Fe, Cr and Ni. For 304L stainless steel, there are the element peaks of Fe, Cr, Ni, Mn and O on the surface, indicating that the corrosion products should be correlated with the oxides and hydroxides of Fe, Cr, Ni and Mn. Furthermore, our recent research results of XPS (the following figure) indicate that the main corrosion products of SiN or 304L stainless steel are the oxides and hydroxides of Fe and Cr, which is in line with the result of EDS. Therefore, we consider that the results of EDS can clarify the composition of oxide films of SiN and 304L stainless steels on the whole.

Figure. XPS results of SiN steel after immersion in nitric acid for 48 h at 95 oC.

Comment 6: Why doesn't the error bar appear in Figure 12? Please complete.

Reply: Thanks for your careful reading. We have added the error bars in Figure 12 in the new version.

Comment 7: The equations from Figures 3, 9 and 12 are not visible. Please correct.

Reply: According to your suggestion, we have redrew these three figures to make the equations clear in the latest version.

Comment 8: The authors should compare their results with other existing references in the literature.

Reply: As required, we have cited relevant references and made a comparative analysis in the new manuscript. Actually, the three-phase corrosion study of stainless steels in high-temperature nitric acid was rarely reported. Even though, we can also find some corrosion data of stainless steels in nitric acid liquid phase [1-3]. By comparison, we can carry out that the SiN stainless steel displays more superior corrosion resistance than other stainless steels in nitric acid liquid phase with oxidizing ions.

[1] R Robin et al., Correlation between composition of passive layer and corrosion behavior of high Si-containing austenitic stainless steels in nitric acid. J. Nucl. Mater. 2008, 375, 65-71, doi:10.1016/j.jnucmat.2007.10.016

[2] B Laurent et al., Silicon enrichment of an austenitic stainless steel - impact on electrochemical behavior in concentrated nitric acid with oxidizing ions, Electrochim. Acta 322 (2019) 134703. https://doi.org/10.1016/j.electacta.2019.134703

[3] B Laurent et al., The kinetics of transpassive dissolution chemistry of stainless steels in nitric acid: the impact of Si, Electrochim. Acta 258 (2017) 653-661. https://doi.org/10.1016/j.electacta.2017.11.110

Comment 9: Why was EDS analysis not performed for each type of corrosion studied? Both EDS and XRD analyses should be completed in the manuscript.

Reply: Thanks for your comment. As for the EDS analysis, we actually obtained the results of EDS in different phases of nitric acid, and we found that the oxide film composition of SiN stainless steel or 304L stainless steel in different phases of nitric acid were similar, so in the manuscript we just introduced the liquid phase EDS results in detail and the oxide film composition after corroding in vapor phase and condensate phase of nitric acid was descripted briefly. With regard to the EDS results, please see the reply to Comment 5.

Comment 10: Some of the bibliographic references have errors. I recommend checking them carefully. For example. Ref. 20, 21.

Reply: Thanks for your advice. We have corrected it in the new version of the manuscript.

It is our hope that our responses to your comments and modifications on the manuscript can be approved by you. Thank you for the opportunity.

Sincerely,

L.M. Zhang

Associate Professor

CAS Key Laboratory of Nuclear Materials and Safety Assessment

Institute of Metal Research, CAS

Round 2

Reviewer 2 Report

The authors have made most of the required corrections and additions. I also understand the explanations given as to why XRD analysis cannot be done.
Regarding point 3 of the previous review, "I don't understand the unit of measurement for corrosion rate "mm/a". The explanations given that mm/a means every year, where "a" refers to "a year"; I do not agree. The unit of measurement is mm/year. Other variants are not electrochemically acceptable.

Author Response

Dear Reviewer:

    Our reply to your comment is as follows.

Comment 1: Regarding point 3 of the previous review, "I don't understand the unit of measurement for corrosion rate "mm/a". The explanations given that mm/a means every year, where "a" refers to "a year"; I do not agree. The unit of measurement is mm/year. Other variants are not electrochemically acceptable.

Reply: Thanks for your comment. We are sorry for the incorrect explanation of "mm/a". The unit of "mm/a" is one of the commonly used units of corrosion rate [1-3]. It indeed means the annual thickness reduction of materials, where "a" we guess refers to "annum" after consulting several professionals. Thank you for pointing out our mistakes. We have revised the explanation of "mm/a" in the new manuscript.

[1] M Takeuchi et al., Corrosion Study of Titanium-5% Tantalum Alloy in Hot Nitric Acid Condensate. In Proceedings of the International Conference on Nuclear Engineering, 2012; pp. 335-340.

[2] AR Shankar et al., Refractory metal coatings on titanium to improve corrosion resistance in nitric acid medium. 2013, 235, 155-164.

[3] Y Sano et al., Effect of metal ions in a heated nitric acid solution on the corrosion behavior of a titanium–5% tantalum alloy in the hot nitric acid condensate. J. Nucl. Mater. 2013, 432, 475-481, doi:10.1016/j.jnucmat.2012.08.009.

It is our hope that our responses to your comments and modifications on the manuscript can be approved by you. Thank you for the opportunity.

Sincerely,

L.M. Zhang

Associate Professor

CAS Key Laboratory of Nuclear Materials and Safety Assessment

Institute of Metal Research, CAS

62 Wencui Road, Shenyang 110016, China

Tel: +86 024-23915904, +86 17609864684

Email: lmzhang14s@imr.ac.cn
